# 3DPortCityMeasure: Methodology for the Comparative Study of Good Practices in Port–City Integration

**María J. Andrade [1,2,\*], João Pedro Costa [2]** 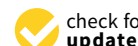 **and José Blasco López [3]**

1   Departamento Arte y Arquitectura, Escuela Tecnica Superior de Arquitectura, Universidad de Málaga, Plaza El Ejido, 2, 29013 Málaga, Spain
2   CIAUD, Centro de Investigação em Arquitetura, Urbanismo e Design, Faculdade de Arquitetura, Universidade de Lisboa, Rua Sá Nogueira, Pólo Universitário do Alto da Ajuda, 1349-063 Lisboa, Portugal; jpc@fa.ulisboa.pt
3   Avda. de la Aurora, 15, 29002 Málaga, Spain; joseblasco@coamalaga.es
\*   Correspondence: mjandrade@uma.es

**Abstract:** Most port cities have a long history of investment in the waterfront, adapting these spaces to improve the quality of life of its inhabitants and increase the tourist interest of the city, in a 50-year process of waterfront regeneration that started in the late 1960s. Even though one of the drivers of development in today's port cities continues to be the transfer of knowledge and experiences between different cases, not all these cities have achieved their goals, nor have all done so in a sustainable way. This article exposes a new methodology, motivated by the need to carry out a comparative study of good practices of port–city integration for twelve specific cases. To enable a comparison of intangible realities such as port–city integration, it is mandatory to have a common benchmark to quantify features of cities from different cases. The 3DPortCityMeasure methodology is intended to provide a framework for analysing port-city integration, with results that supply an immediate understanding of each case. This tool enables direct comparative evaluation and provides support for land use planning and urban design approaches. The results show that the proposed approach for measuring intangible factors in the field of the port–city relationship is a very useful tool, novel in this discipline, and fully applicable to other cases and other urban issues.

**Keywords:** port-city integration; urban studies methodology; comparative study; good practices; sustainability waterfront

## 1. Introduction

Over the last half century, port cities have undergone major changes that, in many cases, have modified the course of their future development. Advancements in technology and the means of production have had major spatial consequences, generating new location dynamics in many ports, and leading to the relocation of modern port areas. For example, the introduction of containers, logistics management, and changes in energy use and shipping routes and networks have freed up older port spaces [1–6]. The evidence seems clear: with each new industrial cycle, the port spaces of the previous cycle become redundant and are freed up. This results in the development of new modern port spaces and the urban or economic renewal of the previous ones in a process that is not immediate but takes place over time [7]. These dynamics of the transformation of old port spaces for new urban uses is what has come to be known since the 1990s as the Waterfront [8–15].

Port cities have regenerated their waterfronts in recent decades through profound initiatives for urban renewal and, since each port has its own special characteristics, this has given rise to a specialist

body of literature. Knowledge in this field is transferred by studying different cases and, since port cities are connected and have similar features, having faced common global challenges throughout history, usually arising from changes to shipping practices, the solutions have been passed on from one port to another [16].

In many cases, the transformation of disused docks has changed the course of the city's future development, but there have also been failures. Therefore, it is essential to learn from the strategies and actions that have improved the quality of life of the inhabitants by increasing the economy and local well-being, compared to many others that have not been so successful.

## 2. Background: City and Port Studies

Port renovation operations began in the 1960s and 1970s in the USA, and since these first interventions up to the present, numerous old port waterfronts have undergone transformations, going through different stages, uses and strategies, successes and failures. In many cases, they have been urban and economic successes, finding new uses for areas that had been abandoned or underutilized and difficult to access even when located at the heart of a city, with diversified programs ranging from residential to tertiary, public amenities or tourism.

However, more than a few have failed, due to the widespread adoption of standardized patterns of intervention, limited to exclusively tertiary sector uses, with a poor understanding of urban needs, and failing to achieve the integration and balance that a city needs from these spaces [16].

In response to the widespread overuse of the American waterfront model, there was a series of studies about the conversions that had been very profitable and successful. As theory has developed from practice, the research projects and studies about each case have proved very valuable for learning not only from successes but also from failures [17–24]. One of the first key factors derived from the research was to understand that continuity with the urban area is essential [25,26]. Another key factor, notable for its absence, was the importance of maintaining the existing urban fabric, as was the need to blend different uses together [27].

These experiences led to a third key factor, very much influenced by competitive rivalry between cities, as transformation processes evolved into real urban marketing strategies. At first, there was a real struggle to attract international events and to get into the rankings of global cities. Later, however, to differentiate themselves from each other, port cities began to focus on the local character and identity of each place, so that their actions paid special attention to the physical, social and cultural characteristics of each local context [28,29].

Several authors have contributed theoretical recommendations that emerged as a result of practices they carried out, making direct reference to trends in thinking in the preceding years that have influenced subsequent generations of waterfront reconversions [18,24,30–36].

The urban and/or productive renewal of the port fabric remains a topical issue, as do the related studies and analyses carried out. The integration of city and port is particularly stimulating for the formation of a rich array of research groups and associations that study and analyse the factors that impact successes and failures, and which mediate between different arms of government to encourage understanding and cooperation. Therefore, determining the port–city relationship and identifying the mode of port–city interaction are difficult yet popular topics in research on port cities [37]. A growing number of universities in port cities are working on research into the port–city relationship, which has become accepted as a field in its own right.

All these reflections about the port city relationship that focus more on the societal factor than the economical one have been compiled into various manuals of good practices, such as *The Cool Sea. Waterfront Communities Project Toolkits* [38] or *Code of Practice on Societal Integration of Ports* by ESPO—European Sea Ports Organisation [39], or even *Plan the City with the Port*, *guide of good practices* by AIVP [40]—International Association Cities and Ports, all of them focusing on strategies to enable 'soft' port activities [41] to coexist with the life of the city. The importance of these has been validated and demonstrated by subsequent studies, which each add their own, new ideas, while maintaining the previous ones [42,43].

As for studies focused more specifically on the economic development of the port, the extensive review of research conducted by Guo [37] differentiated five major research areas in the field of the port–city relationship, summarised as follows: the first area focuses on the spatial characteristics and evolution stages of the relationship between the location of ports and cities; the second area concentrates on the industrial development of the port and its impact on the region; the third one involves the connection between port and city networks, focused on container transport; the fourth area is focused on ecological ports; and the fifth area quantitatively measures the port–city relationship from a regional perspective.

The port city does not refer to a specific scientific category or methodology, because of the diversity of port–city issues and the usual separation of port and urban studies [43,44]. Port–city relationships are more developed in regional studies but few of them have a methodological framework. In fact, port–city relationships are more of a qualitative issue, but a number of authors have proposed some indicators to allow for a comparative approach. Focusing on this fifth area, this type of research is represented by Vallega [45], who established the relative concentration index (RCI) to reveal how Mediterranean port regions and their related human settlements are organized by simply dividing the regional share of throughput by the regional share of population. Most of the studies propose, amend and apply the RCI model, such as Ducruet and Lee [46], who propose applying the RCI to determine the type of relation between port and urban functions by examining the correlation between city population and container throughput, or the DCI (dynamic concentration index) model proposed by Guo [37], and others develop new methodologies, such as Schiper et al. [47], that involve ranking various long-term port plans and port vision documents against a set of social, economic, and environmental key performance indicators (KPIs).

All these studies that quantitatively measure the port–city relationship are focused on an economical perspective [37,44–49]. However, the economic function of ports can only be sustained in the longer run if the societal function is taken seriously [39].

It is precisely in this societal perspective, in the human factor, that this paper is located. As Ducruet explains, the lack of data on port and urban functions has restrained comparative studies [49]. As more ports are transformed, more cases studied and more projects planned, there is an increasing need for benchmarks to enable comparative evaluations backed up by indicators that are effective and easy to obtain. Despite the increasing number of studies on waterfront transformations [17–24], focused on obtaining a success criteria that can be applied in future interventions, there is no common methodology. It seems that this diversity of initiatives has often resulted in international debates based on the exposure of individual local cases without an agreement on common research strategies on the phenomenon. There is a need for a methodology that can provide a clear and quantitative vision of the degree of port–city integration achieved by waterfront transformations, one that can be used to analyse different cases to arrive at a comparison.

This paper proposes a new methodology that enables us to use a common benchmark to quantify port–city integration in different cases, compare them, and arrive at a global overview of the relationship between city and port. This novel methodology is part of an international research project on the comparative study of port–city integration carried out in twelve cities [18].

The difficulty of such a societal approach is the systematic measurement of a complex phenomenon which is not well-defined in the literature.

## 3. 3DPortCityMeasure: A Methodology for the Comparative Study of Good Practices in Port-City Integration

The 3DPortCityMeasure methodology seeks to contribute to the scientific literature by proposing a framework for understanding and quantifying the configurations of urban systems of a waterfront from the perspective of port–city integration. The proposed framework can be used at different spatial scales and is a very useful tool that can be completely extrapolated not only to other cases in the same

scope, but to any comparative study of urban issues relating to edge spaces, or to transition between two realities, such as city–infrastructure or city–nature.

This methodology starts with an exhaustive analysis of the most relevant waterfront interventions throughout history in different parts of the world to obtain three main interactions that characterize them, three dimensions, each in turn comprising a series of indicators that quantify this interaction, and which form the basis of this methodology. Bohanec et al. [50] suggest that the most difficult part of developing the model design is the identification of the model structure and elements, as it is defined manually in a lengthy process based upon knowledge about the problem domain.

The lessons learned from this type of successful experience would be useful as recovery strategic models for other cities that have not yet taken the appropriate measures or in which the measures developed have not had the expected success. This would apply for as long as the appropriate comparative framework was provided [51].

The methodology takes port–city integration to be the interaction between these two realities at the same level, with no hierarchy, in a bidirectional relationship, characterized by three dimensions that derive chronologically from the different stages of waterfront evolution [52]. They are the characteristic dimensions of waterfronts throughout history, and those which current studies continue to focus on: the physical dimension (PD) (structural), which arises from the early American projects; the functional dimension (FD) (flow), which arises from European experiences, following on from the single-function tertiary character of early American experiences; the social dimension (SD) (in both the human and environmental factors), which arises from the most recent projects that strive to enhance a city's port identity by maintaining port activities in historic ports. It is important to break down the concept of integration into these three dimensions to avoid the vagueness that otherwise prevents quantification.

## 3.1. Collection of Cases

A set of indicators is proposed to describe interaction patterns in accordance with the three main dimensions. These evaluation indicators come from a literature review on good practice in port cities [38–40], and the analysis of the different cases of interest (see Table 1) that have been part of the research.

The port cities analysed are twelve European port cities that have transformed their waterfronts and maintained port activity in their docks, either with cruise liners or container ships. The closeness of the shipping activity (both container vessels and cruise liners) to the historic city centre, and the acceptance of these activities by residents, make these cases very relevant as regards the reality of harmonious coexistence between a working port and its city. These cases are very different, not only in scale, morphology and use, but also in the port's location in the city. However, they have found huge social acceptance and have greatly improved the urban quality of their cities. They are examples of sustainability and have maintained the identity of the port in the waterfront.

Each one stands out for its application of a good practice. Each good practice constitutes one of the indicators that are grouped into the three large integration blocks considered. They stand out in their use of the majority of the good practices, too, resulting in actions designed by and for the citizen.

The cases of port cities analysed are, from the northern coastline, Vigo, Gijon, Santander and Hamburg and, from the Mediterranean coast, Malaga, Valencia, Barcelona, Marseille and Genoa, which are all examples of transformations of historic ports. The more advanced examples, Oslo, Copenhagen and Helsinki, have transformed not only their historic ports but also their container ports.

## 3.2. Description and Values of the Three Dimensions and Their Indicators for the Quantification of Port–City Integration

Applying this set of indicators is fundamental for capturing the level of port–city integration, since there is, in our opinion, no single measure that can provide this information about a reality that is so complex, even intangible. As indicated in Tables 2 and 3, each of these 12 indicators, four in every dimension, can be assigned a score from 1 to 5: Value 1 in the worst-case definition, and 5 in the best-case definition.

**Table 1.** Summary of successful waterfront case studies (2012).

| Case Collection | Population | Cruisers | Containers | Dominant Success | Initiative Profile | Web Reference Data |
|---|---|---|---|---|---|---|
| 1. Vigo (SP) | 297,241 | 233,644 | 572,784 | Physical Continuity | Physical Dimension, PD | www.apvigo.com<br>www.fupv.es<br>www.hoxe.vigo.org<br>www.uvigo.es |
| 2. Oslo (NW) | 607,292 | 310,000 | 201,893 | Continuity of Movement | Physical Dimension, PD | www.oslohavn.no<br>www.ohv.oslo.no<br>oslo.kommune.no<br>www.uio.no |
| 3. Barcelona (SP) | 1,619,337 | 2,347,976 | 1,900,000 | Visual Continuity | Physical Dimension, PD | www.bcn.es<br>www.bcnregional.com<br>www.consorcielfar.org<br>www.portdebarcelona.es<br>www.portvellbcn.com<br>www.ub.edu/es |
| 4. Copenhagen (DK) | 1,080,000 | 662,000 | 153,000 | Architectural Quality: Planning, Buildings and Public Spaces | Physical Dimension, PD | www.cmport.com<br>www.byoghavn.dk<br>https://byoghavn.dk/nordhavn/<br>www.ku.dk |
| 5. Genoa (IT) | 610,830 | 860,290 | 1,758,858 | Blend of Uses | Functional Dimension, FD | www.smart.comune.genova.it<br>www.portsofgenoa.com<br>www.comune.genova.it<br>www.genoaportcenter.net<br>www.unige.it |
| 6. Malaga (SP) | 568,030 | 638,845 | 476,997 | Balance of Public Spaces | Functional Dimension, FD | www.malagaport.com<br>www.malagaturismo.com<br>www.muelleuno.com<br>www.malaga.eu<br>www.omau-malaga.com<br>www.puertomalaga.com<br>www.uma.es |
| 7. Santander (SP) | 181,589 | 7457 | 1520 | Location of Focal Uses | Functional Dimension, FD | www.puertosantander.es<br>cantabriacampusinternacional.com<br>www.rfev.es<br>www.fcvela.com<br>www.santander2014.com<br>www.ayto-santander.es<br>www.uimp.es<br>www.unican.es |

**Table 1.** *Cont.*

| Case Collection | Population | Cruisers | Containers | Dominant Success | Initiative Profile | Web Reference Data |
|---|---|---|---|---|---|---|
| 8. Valencia (SP) | 798,033 | 253,743 | 4,327,371 | Flexibility | Functional Dimension, FD | www.valencia.es<br>www.valenciaport.es<br>www.uv.es |
| 9. Hamburg (DE) | 1,770,000 | 245,700 | 8,000,000 | Environmental Integration | Social Dimension, SD | www.elbphilharmonie.de<br>www.hamburg.de<br>www.hafen-hamburg.de<br>www.hafencity.com<br>www.hcu-hamburg.de<br>www.uni-hamburg.de |
| 10. Helsinki (FI) | 588,549 | 360,000 | 400,000 | Public Image | Social Dimension, SD | www.hel.fi/hki/taske/en/Urban+Development<br>www.helsinki.fi<br>www.portofhelsinki.fi<br>www.southharbour.fi |
| 11. Gijon (SP) | 277,559 | 5082 | 318,065 | Education and Employment | Social Dimension, SD | www.gijon.es<br>www.puertogijon.es<br>www.puertodeportivogijon.es |
| 12. Marseille (FR) | 859,543 | 1,741,282 | 953,435 | Port in Local People's Daily Life | Social Dimension, SD | www.marseille-port.fr<br>www.euromediterranee.fr<br>www.reseau-euromed.org/en/institut-de-mediterranee/<br>www.univ-provence.fr |

**Table 2.** 3DPortCityMeasure: Description of indicators and values of the three dimensions for the quantification of port–city integration.

| Dimension | Indicator | Rank | Definition |
|---|---|---|---|
| | 1. Physical Continuity: city–port access and connections. Ground-level road/rail infrastructure barriers. | 5 | Complete pedestrian continuity. No barriers. No ground-level road traffic. |
| | | 4 | Complete pedestrian continuity. Moderate ground-level road traffic (ground-level public transport and residents). |
| | | 3 | Barriers (road traffic) with continuous pedestrian access points. |
| | | 2 | Barriers with temporary access points (pedestrian access). |
| | | 1 | Barriers (roads, port fences, railway tracks…). |

**Table 2.** *Cont.*

| Dimension | Indicator | Rank | Definition |
|---|---|---|---|
| PHYSICAL DIMENSION PD | 2. Continuity of Movement: Town connected to sea by development of transport and patterns of movement. Intermodal, sustainable transport network, in which land transport complements maritime (blue) transport. | 5 | Blue transport is part of the city's public transportation network. Intermodal. |
| | | 4 | Terrestrial public transport network reaches into port area to allow itineraries in different modes: pedestrian, bicycle, tramways... |
| | | 3 | Terrestrial public transport network reaches into port area: local people can access the port, and cruise passengers the town. |
| | | 2 | Terrestrial public transport network reaches into port–town border area. |
| | | 1 | No public transport near the port area. |
| | 3. Visual Continuity: Visual permeability, presence of the port in the town, places for viewing port activities, port atmosphere in the town (businesses, shops...) | 5 | Complete visual connection with sea, docks, activities: viewpoints, streets, topography... |
| | | 4 | Different points for observing the port and its activities. |
| | | 3 | Views of some port activity from town, port atmosphere in the vicinity (shops, businesses...) |
| | | 2 | Only occasional views of some port activity (view of cranes from street, cruise ship from some point...) |
| | | 1 | No visual permeability between the port and the city. |
| | 4. Architectural Quality: Of both town planning (well-grounded and responding to the problems of the local context), and of buildings and public spaces. | 5 | Good quality planning with long-term, global vision. Good quality public spaces and buildings. Heritage conservation and management. |
| | | 4 | Planning with a comprehensive vision. New buildings that create urban environment. Conservation of some heritage buildings. Quality public spaces. |
| | | 3 | Planning with a comprehensive vision. New buildings that create urban environment. No conservation of heritage buildings. Quality public spaces. |
| | | 2 | No comprehensive vision. Buildings that do not create urban environment. Quality public spaces |
| | | 1 | No comprehensive vision. Buildings that do not create urban environment, public spaces are small or poor quality. |
| | 5. Blend of uses: Urban complexity; balance between town and port uses, local and global uses, daily and occasional uses, uses by tourists and by locals. | 5 | Blend of uses. Prevalence of port uses, marine-related uses, everyday urban use, leisure and entertainment use (fish market, university, transport, offices...). |
| | | 4 | Dockside uses. Urban cultural and business uses, and daily uses (transport, offices, housing, university...). |
| | | 3 | Dockside uses. Urban cultural and business uses (museum, auditorium, shopping centre, restaurants. Docks: marina). |
| | | 2 | Urban cultural and business uses are mixed. No dockside uses. |
| | | 1 | Single-function. Leisure and entertainment use only. Shopping centres. No dockside uses. |

**Table 2.** *Cont.*

| Dimension | Indicator | Rank | Definition |
|---|---|---|---|
| FUCTIONAL DIMENSION FD | 6. Balance of Public Spaces: Preservation of the characteristic openness of port spaces. Building density on quaysides. Berth density in docks. | 5 | Balance between low-density and free spaces both in docks and quays. Places for recreational walking and other activities, open spaces. |
| | | 4 | Public space. Medium quayside density. Balanced docks. |
| | | 3 | Public space. Medium quayside density. High berth density in docks. |
| | | 2 | Few public spaces. Medium quayside density. High berth density in docks. |
| | | 1 | Few public spaces. High building density on quaysides. High berth density in docks. |
| | 7. Focal Uses Location: Uses that form part of the local inhabitants' daily activities take place at the docks, beyond the leisure area. | 5 | Uses that complement each other are distributed between port and town. Daily, everyday uses. Use of heritage sites. |
| | | 4 | Everyday uses, such as offices, university, market, are located on the quays... |
| | | 3 | Buildings aimed at activities such as leisure, shops, and restaurants, are located on the quay. |
| | | 2 | Buildings for occasional use (such as auditorium, convention centre) are located on the quay. |
| | | 1 | Buildings that hosted large events, now unused, located on the quay. |
| | 8. Flexibility: Ability to maintain the multi-purpose nature of these places throughout history. Ability to adapt to demand at all times. | 5 | Short leases. A few new non-revertible but multi-purpose buildings. Low density on quays, and complete flexibility of public space. |
| | | 4 | Some new buildings are non-revertible but multi-purpose. Medium-term leases. Low density that allows complete flexibility of public space. |
| | | 3 | Non-revertible buildings. Long leases. Medium/low density allowing flexible use of public space. |
| | | 2 | Non-revertible buildings. Medium/low building density. Long leases. |
| | | 1 | Non-revertible buildings. High building density. Buildings with very inflexible spaces (housing); very long leases. |
| | 9. Environmental Integration: Monitoring of the port's environmental aspects, solving or minimizing pollution problems arising from an active port being located in an urban area. Port space for power generation. | 5 | Energy production. Pro-active environmental management to reduce port's environmental impact. Disclosure of measurements and data. |
| | | 4 | Pro-active environmental management to reduce port's environmental impacts, such as water and air quality, noise, refuse. Disclosure of measurements and data. |
| | | 3 | Environmental management. Quality Control. Clean Port. Little disclosure. |
| | | 2 | Environmental management. Medium level of monitoring, surveillance and control. |
| | | 1 | Minimal environmental management. Little monitoring. |

**Table 2.** *Cont.*

| Dimension | Indicator | Rank | Definition |
|---|---|---|---|
| SOCIAL DIMENSION SD | 10. Public Image: Actions and initiatives to (1) understand public perception of the port; (2) restore the port's reputation through visits and specific events and/or (3) exhibition and visitor information centres, port museums … | 5 | Public perception of the port by local inhabitants is understood (surveys, social media, etc.). Visits and events to attract people to the port. Port visitor information centres, museums… |
| | | 4 | Actions to fulfil at least two of the three aspects (understanding public perception, one-off or continuous visitor attraction actions). Clear web site. High social network activity. |
| | | 3 | Actions to fulfil at least two of the three aspects (understanding public perception, one-off or continuous visitor attraction actions). Clear web site. Moderate social network activity. |
| | | 2 | Negative public perception. Actions are few (more than three), sporadic, casual, not part of any strategy. Confusing web site. Little social network activity. |
| | | 1 | Negative public perception. Actions are very few (one to three), sporadic, casual, not part of any strategy. Very confusing web site. No social network activity. |
| | 11. Education and Employment: Educational programmes at different levels, in schools, vocational training, and university education, to inform and train students about different port, maritime and logistics jobs. | 5 | Outreach about port activities in different academic contexts (visits from schools, study programmes...), provision of training and research programmes in the port, maritime and logistics sectors. |
| | | 4 | Training of port workers is encouraged, there are visits from schools to the port, and some provision of marine-related university courses such as Marine Sciences or Marine Engineering. |
| | | 3 | Training of port workers is encouraged, and there are visits from schools to the port. |
| | | 2 | Training of port workers is encouraged, but there is no educational provision outside the port |
| | | 1 | No academic or training provision either for existing workers or for new staff. |
| | 12. Port in Local People's Daily Life: As regards mobility and public transport, as a place of study or work, place of residence, for shopping, as a market, fish market, place of leisure, for walking, for children's games, for contemplation... | 5 | Direct integration of the port space in the population's everyday activities (transport, work, residence, market...). |
| | | 4 | The port relates to some of the inhabitants' everyday activities. |
| | | 3 | The port relates to the inhabitants, not in their everyday activities, but for leisure (walking, shopping, restaurants...) |
| | | 2 | The port relates to a minority of the inhabitants (wholesale fish market, auditorium, congress centre, etc.). |
| | | 1 | The port has no relation to the inhabitants' everyday activities. |

**Table 3.** Values of indicators.

| | Indicator | Rank | Value |
|---|---|---|---|
| **Physical Dimension (PD)** | PDc | 5 | $B = 0\ Pap = 3$ |
| | | 4 | $B = 1\ Pap = 3$ |
| | | 3 | $B = 2\ Pap = 2$ |
| | | 2 | $B = 2\ Pap = 1$ |
| | | 1 | $B = 2\ Pap = 0$ |
| | PDm | 5 | $PTm = 1\ PTt = 3$ |
| | | 4 | $PTm = 0\ PTt = 3$ |
| | | 3 | $PTm = 0\ PTt = 2$ |
| | | 2 | $PTm = 0\ PTt = 1$ |
| | | 1 | $PTm = 0\ PTt = 0$ |
| | PDv | 5 | $Vp = 4\ Va = 2$ |
| | | 4 | $Vp = 3\ Va = 1$ |
| | | 3 | $Vp = 2\ Va = 1$ |
| | | 2 | $Vp = 1\ Va = 0$ |
| | | 1 | $Vp = 0\ Va = 0$ |
| | PDq | 5 | $Qp = 3\ Qb = 4\ Qps = 1$ |
| | | 4 | $Qp = 2\ Qb = 3\ Qps = 1$ |
| | | 3 | $Qp = 2\ Qb = 2\ Qps = 1$ |
| | | 2 | $Qp = 1\ Qb = 1\ Qps = 1$ |
| | | 1 | $Qp = 1\ Qb = 1\ Qps = 0$ |
| **Functional Dimension (FD)** | FDu | 5 | $Ud = 2\ Uu = 2$ |
| | | 4 | $Ud = 1\ Uu = 2$ |
| | | 3 | $Ud = 1\ Uu = 1$ |
| | | 2 | $Ud = 0\ Uu = 1$ |
| | | 1 | $Ud = 0\ Uu = 0$ |
| | FDps | 5 | $Pss = 1\ Psd = 1\ Psb = 2$ |
| | | 4 | $Pss = 1\ Psd = 1\ Psb = 1$ |
| | | 3 | $Pss = 1\ Psd = 1\ Psb = 0$ |
| | | 2 | $Pss = 0\ Psd = 1\ Psb = 0$ |
| | | 1 | $Pss = 0\ Psd = 0\ Psb = 0$ |
| | FDfu | 5 | $Fub = 5\ Fc = 1$ |
| | | 4 | $Fub = 4\ Fc = 1$ |
| | | 3 | $Fub = 3\ Fc = 0$ |
| | | 2 | $Fub = 2\ Fc = 0$ |
| | | 1 | $Fub = 1\ Fc = 0$ |
| | FDfl | 5 | $Fl = 5–10$ years $Flr = 1\ Fld = 2$ |
| | | 4 | $Fl = 10–25$ years $Flr = 1\ Fld = 2$ |
| | | 3 | $Fl = 10–25$ years $Flr = 0\ Fld = 1$ |
| | | 2 | $Fl = 25–50$ years $Flr = 0\ Fld = 1$ |
| | | 1 | $Fl = 25–50$ years $Flr = 0\ Fld = 0$ |
| **Social Dimension (SD)** | SDe | 5 | $Em = 2\ Emo = 2\ Ep = 1$ |
| | | 4 | $Em = 2\ Emo = 2\ Ep = 0$ |
| | | 3 | $Em = 1\ Emo = 2\ Ep = 0$ |
| | | 2 | $Em = 1\ Emo = 1\ Ep = 0$ |
| | | 1 | $Em = 0\ Emo = 0\ Ep = 0$ |
| | SDps | 5 | $Pp = 1\ Pa = 3\ Pws = 3$ |
| | | 4 | $Pp = 1\ Pa = 2\ Pws = 3$ |
| | | 3 | $Pp = 1\ Pa = 2\ Pws = 2$ |
| | | 2 | $Pp = 0\ Pa = 1\ Pws = 1$ |
| | | 1 | $Pp = 0\ Pa = 0\ Pws = 0$ |
| | SDee | 5 | $Ep = 2\ Et = 1\ Ei = 1$ |
| | | 4 | $Ep = 1\ Et = 1\ Ei = 1$ |
| | | 3 | $Ep = 0\ Et = 1\ Ei = 1$ |
| | | 2 | $Ep = 0\ Et = 1\ Ei = 0$ |
| | | 1 | $Ep = 0\ Et = 0\ Ei = 0$ |
| | SDpc | 5 | $Pct = 1\ Pcd = 2\ Pcl = 1$ |
| | | 4 | $Pct = 1\ Pcd = 1\ Pcl = 1$ |
| | | 3 | $Pct = 0\ Pcd = 1\ Pcl = 1$ |
| | | 2 | $Pct = 0\ Pcd = 0\ Pcl = 1$ |
| | | 1 | $Pct = 0\ Pcd = 0\ Pcl = 0$ |

The input data (attributes of successful cases in terms of port–city relationship) were collected from mostly theoretical academic studies, which are the most common in this field, and data from port, city and university websites of each case.

After quantifying each indicator in every single case, diagrams are provided (see Figure 4) to illustrate the proposed framework, show more clearly the different cases, and allow a comparative visual analysis. In this way, we intend to provide a framework for analysis and for obtaining a comprehensive view of the quality of a port–city integration—a tool to enable direct, comparative study.

3.2.1. Physical Dimension (PD)

The first of the three proposed spatial interaction dimensions, the Physical Dimension (PD), refers to the structure of the system. While physical integration is not the most important aspect of the port–city relationship, it is fundamental to it. As a basic factor in the integration process, this has been present since the earliest experiences, American projects, which, although they went too far with leisure and entertainment-oriented tertiary amenities, did solve the issues around physical connections between the new projects and their cities.

PD is a complex reality composed of various connection factors. The four indicators that contribute to describing this dimension are:

1.  Physical Continuity (PDc) measures the continuity of public space and of the urban fabric. This is determined by the barriers ($B$)—from $B = 0$, for continuity without barriers, $B = 1$, indicating moderate ground-level road traffic, and $B = 2$ for barriers such as roads, port fences and railway tracks—and by the type of pedestrian access point ($Pap$)—$Pap = 3$ for complete pedestrian continuity, $Pap = 2$ for continuous pedestrian access points, $Pap = 1$ for more than three temporary pedestrian access points, and 0 for less than two temporary pedestrian access points.

    As can be observed in Tables 2 and 3, a PDc = 1 indicates barriers (roads, port fences, railway tracks) in the total interface and less than two temporary pedestrian access points; PDc = 3 refers to barriers with permanent access points (e.g., an overpass); and a PDc = 5 is complete pedestrian continuity, with neither barriers nor ground-level road traffic;

2.  Continuity of Movement (PDm) and the reach of public transport networks into these new spaces: in this factor, PDm = 1 means no terrestrial public transport ($PTt = 0$) or maritime ($PTm = 0$) near the port area; a port city with PDm = 2 has terrestrial public transport ($PTt = 1$) just into the port–city border area, while PDm = 3 indicates that the public transport ($PTt = 2$) network reaches into the port area and a PDm = 4 indicates that it also allows itineraries in different modes ($PTt = 3$), finally, a PDm = 5 is when not only the terrestrial public transport ($PTt = 2$) network reaches into the port area, but blue (maritime) transport ($PTm = 1$) is also part of the city's public transportation network;

3.  Visual Continuity (PDv) from the city's streets and viewpoints: the perception of port activity by the city reinforces the feeling of acceptance. Two parameters are used for the evaluation, the view of the port ($Vp$), and the port atmosphere in the city ($Va$), for example, in store windows. No visual permeability between the port and the city ($Vp = 0$) and any port atmosphere ($Va = 0$) is evaluated with PDv = 1. There are still ports with a large wall that separates them from the city not only physically but also visually. A PDv = 2 indicates that there are occasional views of some port activity ($Vp = 1$), e.g., views of cranes from a street or views of a cruise ship from some point, but without any port atmosphere ($Va = 0$), while PDv = 3 means not only some views of port activity ($Vp = 2$), but also a port atmosphere in the vicinity (shops, businesses...) ($Va = 1$), and PDv = 4 means different points for observing the port and its activities ($Vp = 3$), and port atmosphere in the vicinity ($Va = 1$), PDv = 5 indicates a complete visual connection with the sea, docks, activities ($Vp = 4$), and a port atmosphere in the city ($Va = 2$);

4.  Architectural Quality (PDq) of both town planning ($Qp$) (well-grounded and responding to the problems of the local context), and of buildings ($Qb$) and public spaces ($Qps$).

$Qp$ is evaluated by the global and long-term vision (tv): $Qp = 1$, no comprehensive and short-term vision (tv < 4 years), $Qp = 2$, comprehensive and medium-term vision (4 < tv < 12 years), $Qp = 3$, global and long-term vision (tv > 12 years).

One of the main factors that brings identity to a place is the conservation of heritage, and if buildings also generate an urban environment, they will be evaluated with $Qb = 4$, but if only a few aspects of heritage are conserved and buildings generate an urban environment, $Qb = 3$. A $Qb = 2$ means that there are only new buildings, but they create an urban environment, and $Qb = 1$ indicates that there are only new buildings and they do not generate any urban activity around them.

Not only is the urban activity of the buildings important, but also the quality of these public spaces ($Qps$). This is evaluated as $Qps = 0$ without quality and $Qps = 1$ with quality.

As can be observed in Table 3, PDq = 2 means ($Qp = 1$, $Qb = 1$, $Qps = 1$) that there is not a comprehensive and short-term vision, and the buildings are only new ones and do not create an urban environment, but they have quality public spaces. A PDq = 4 means ($Qp = 2$, $Qb = 3$, $Qps = 1$) that planning has a comprehensive and medium-term vision, and new buildings that create an urban environment, with the conservation of some heritage buildings and/or quality public spaces.

Therefore, physical integration goes beyond ground plans, and beyond urban layouts; it becomes a major factor in the human perception of a port city.

### 3.2.2. Functional Dimension (FD)

The second of the three proposed spatial interaction dimensions, the Functional Dimension (FD), refers to system flows. According to Simmons [53], like Limtanakool et al. [54], the degree of integration of a system is a function of 'the sum of all flows of some types within the system as a whole'. Therefore, the types of uses and their strategic locations are relevant to port–city integration, since the more the uses located in the city depend on those located in the port, the more connection networks there will be between them, increasing flows in both directions and thus increasing the presence of the port in the daily activities of the local residents. This generates synergies between city and port, ensuring the durability of the transformed port spaces.

The integration of port and city is achieved not only by ensuring continuity of the urban fabric (PD) but also by introducing a mixture of uses and activities to revitalize the space (FD). The numerous projects that have been done in different countries over the last two decades have been thoroughly studied and written about. Experiences from one project have fed into another, and several models of intervention have been attempted. These range from the early American model, with its predominance of tertiary uses and leisure spaces, up to the present model that stresses a blend of uses, maintaining port activities that can coexist with the city in a sustainable way, through the search for a new balance between these realities.

The four indicators that contribute to describing this dimension are:

5. Blending Uses (FDu) to regenerate areas that are characterized by an interesting complexity, with sea-related uses, activities linked to previous and original uses, and uses related to the working port. It is also necessary to integrate the hard functions of a port, as defined by Van Hooydonk [41]. Maintaining an active port will enhance a city's port identity. In this factor, two values are used, on one hand the dockside uses ($Ud$)—$Ud = 0$ when there is any port activity in the basin, $Ud = 1$ if there is a marina, or $Ud = 2$ with port uses—and, on the other hand, function diversity ($Uu$) is evaluated as $Uu = 0$ when there is only leisure use, $Uu = 1$ if this use is mixed with cultural and business uses, and $Uu = 3$ when these are also mixed with daily urban uses such as fish markets, university, or transport.

   In this indicator, FDu = 1 means single-function, leisure and entertainment use only, shopping centres ($Uu = 0$), and no dockside uses ($Ud = 0$). A score of FDu = 5 is a waterfront with a blend of uses, with a prevalence of port uses, marine-related uses, ($Ud = 2$), everyday urban use, leisure and entertainment use (fish market, university, transport, offices) ($Uu = 2$);

6. Balancing Free Spaces (FDps), which refers to maintaining the proportion of free space that existed in the port with its previous activity, as part of its intangible heritage. It involves avoiding the great temptation for many cities to privatize these spaces and fill them with buildings. Three values are used, first the public space (*Pss*), that is *Pss* = 0 when it is more private than public, and *Pss* = 1 when it is public at all; then, the berth density in docks (*Psdz*), that is evaluated from *Psd* = 0 with a high berth density, to *Psd* = 1 with a balanced dock; and on the other hand, building density (*Psb*), from *Psb* = 0 with a high building density, *Psb* = 1 with a medium quayside density, and Psb = 2 with a low density, and open spaces for recreational walking and other activities.

   For instance, FDps = 2 is a waterfront with few public spaces (*Pss* = 0) and high building density on the quays (*Psb* = 0), but the basin is balanced and there is not a high berth density (*Psd* = 1). An FDps = 4 means public space (*Pss* = 1), medium quayside density (*Psb* = 1), and balanced docks (*Psd* = 1);

7. Focal Uses' Locations are the provision of uses that can mobilize urban dynamics (FDfu), combining everyday uses with those that entail occasional visits in a way that ensures continuous occupation of the dockside. In this factor, two values are used: the use of building on the quay, (*Fub*)—evaluated from *Fub* = 1, in this case where there are buildings that hosted large events, as in many cases on waterfronts, but now are unused, *Fub* = 2 in cases of buildings for occasional use, such as auditoriums, or convention centres, *Fub* = 3 for buildings aimed at activities such as leisure, shops and restaurant, *Fub* = 4 for everyday uses, and *Fub* = 5 for everyday uses in heritage buildings—and the relationship with other uses located at city (*Fc*), as *Fc* = 0 when they are independent, and *Fc* = 1 if there is synergy between them, e.g. the library and the university, or the market and the fish market.

   An FDfu = 1 means that the waterfront has many buildings that hosted large events, now unused, located on the quay (*Fub* = 1) and without any connection with city uses (*Fc* = 0), as opposed to FDfu = 5, which means mixed daily uses (*Fub* = 5) that complement each other are distributed between the port and city (*Fc* = 1), including everyday uses and even use of heritage sites;

8. Flexibility in both spaces and buildings (FDfl) can increase the value of interventions, as it allows them to be adapted to the changing needs of both the port and city, to host multiple events and activities. In this factor, three values are used: lease terms (*Fl*)—a very long lease *Fl* = 25–50 years, does not allow adaptability to the demands and needs of each moment, while a short lease, *Fl* = 5–10 years allows more flexibility, and a medium lease *Fl* = 10–25 years, hinders adaptability—a revertible or flexible use, *Flr* = 1, or an inflexible use such as housing, with *Flr* = 0, and the third value is the building density, that determines the flexible use of public spaces, and could be high (*Fld* = 0), medium (*Fld* = 1) or low (*Fld* = 2).

   An FDfl = 1 means non-revertible buildings, buildings with very inflexible spaces (housing) (*Flr* = 0), a high building density (*Fld* = 0), and very long leases (*Fl* = 25–50 years), while an FDfl = 5 means short leases (*Fl* = 5–10 years), a few new, non-revertible but multi-purpose buildings (*Flr* = 0), low density on the quays, and complete flexibility of public space (*Fld* = 2).

### 3.2.3. Social Dimension (SD)

The third of the three proposed dimensions of spatial interaction is the Social Dimension (SD). The social integration of ports is an essential part of port governance. It refers to actions that the port authorities take to optimize the relationship between the port and its social context, focusing on the human factor and environmental responsibility. The port must take steps to optimize relationships between the port, its environment and its social context. In both the managing of the port and the ongoing process of port–city integration, the authorities must take into consideration a wide and divergent range of opinions and specific objectives of stakeholders such as port workers, neighbourhood associations, involved companies, and planning professionals. The management, promotion and development of a port's soft values [41] are essential tools for achieving social integration, but they need

to be complemented by hard values such as attracting employment through investment in educational infrastructure. Without social integration, satisfactory port–city integration can never be achieved, because in the end only people can really connect the port with urban life.

The four indicators that contribute to describing this dimension are:

9.　Environmental Integration (SDe) is crucial for the coexistence of a port close to the city, for providing inhabitants with well-being in a city that is green, environmentally friendly and balanced, and with a higher environmental quality. This is determined by the environmental management level (*Em*), which can be minimum (*Em* = 0), medium (*Em* = 1) or pro-active (*Em* = 2); the environment monitoring (*Emo*) corresponding to low (*Emo* = 0), medium (*Emo* = 1) or high monitoring (*Emo* = 2); and energy production (*Ep*) such as wind energy, solar energy (*Ep* = 1).

In this factor, a port with SDe = 1 means minimal environmental management and little monitoring, while a port with SDe = 5 indicates high monitoring, energy production and pro-active environmental management to reduce the port's environmental impact;

10.　Public Image and Support (SDps): ports usually have a negative public image. People associate them with pollution and noise, traffic and bad smells, as well as the smuggling of drugs, immigrants and illegal goods. All these factors tend to make the local inhabitants want to stay away. This indicator measures actions and initiatives to understand the public perception of the port (*Pp*), which can be *Pp* = 0 for any action, or *Pp* = 1 for actions such as surveys or social media. Attempts restore the port's reputation through visits and specific events are referred to as (*Pa*): for sporadic attempts, *Pa* = 0, for 1–3 actions, *Pa* = 1, for more than three, *Pa* = 2, and a strategy for continuous visitors means Pa = 3. In addition to the above, there are port visitor information centres, museums, etc., and website and social network activity (*Pws*), in which a confusing website and no social network activity, *Pws* = 0, little social network activity results in a *Pws* = 1, clear website and moderate social network activity means a *Pws* = 2, and a clear website and high social network activity results in *Pws* = 3. A waterfront with a *Pws* = 2 means a negative public perception (*Pp* = 0), where actions are few (more than three), sporadic, casual, and not part of any strategy (*Pa* = 1), with a confusing website and little social network activity (*Pws* = 1);

11.　Education and Employment (SDee): Involving the local residents with the life of the port through education and training affects the development of both the port and the city. This is determined by educational programmes such as university (*Ep*), where a port city with university degrees, courses and research programmes in the port, maritime and logistics sector is a *Ep* = 2, a city with a university degree is a *Ep* = 1, and one without any is *Ep* = 0. If the training of port workers (*Et*) is encouraged, then *Et* = 1, but no training provision results in *Et* = 0. Important information is acquired on the labour activity in the port (*Ei*), such as visits from schools or high schools *Ei* = 1.

A port city SDee = 2 indicates that the training of port workers is encouraged (*Et* = 1), but there is no educational provision outside the port (*Ep* = 0) and any visit from schools (*Ei* = 0), whereas, for instance, an SDee = 4 means that the training of port workers is encouraged (Et = 1), there are visits from schools to the port (*Ei* = 1), and there is some provision of marine-related university courses (*Ep* = 1);

12.　Integrating the Port with the City (SDpc): integrating the port with the city residents' everyday life helps a port city recover its identity. The port becomes part of the inhabitants' daily life again, in aspects such as travel and mobility (*Pct*); as a place of study or work, residence, a retail market place, wholesale fish market, etc., with (*Pcd*) showing this on three levels—*Pcd* = 0 without daily uses, *Pcd* = 1 with 1–3 daily uses and *Pcd* = 2 with more than three daily uses; and as a place of leisure, contemplation and enjoyment (*Pcl*). This reconciliation is crucial for achieving coexistence and reincorporating the port and its activity in inhabitants' daily lives.

An SDpc = 2 expresses that the port relates to a minority of the inhabitants (auditorium, congress centre, etc.), and an SDpc = 5 shows direct integration of the port space in the population's everyday activities (transport, work, residence, market).

## 4. Applying the Framework, A Case Study: Malaga

In this section, the proposed theoretical framework is illustrated using empirical data. The twelve selected cases, from different countries, morphologies, climates and integration statuses, are analysed to demonstrate the usefulness of the proposed tool.

The study of these cities provides a general overview of the intervention criteria of several international initiatives, past and present. This tool greatly enriches the perception of the different cases, enabling us to determine deficiencies and needs in new cases, and to suggest possible solutions that have found great acceptance and success in other places and could be adapted to each local context, taking the existing study as a field reference.

Research has been carried out around multiple realities, expressed in the different cases presented, and this has enabled us to define some common rationales. The complete process was followed, and the details of the investigation are available in Andrade et al. [18]. However, for a better understanding of the process, the application of the methodology in one of the case studies, Malaga (Figures 1 and 2), is detailed below, at two different times (2012–2019), to demonstrate in turn the application and usefulness of the tool in local self-assessment (see Table 4 and Figure 3).

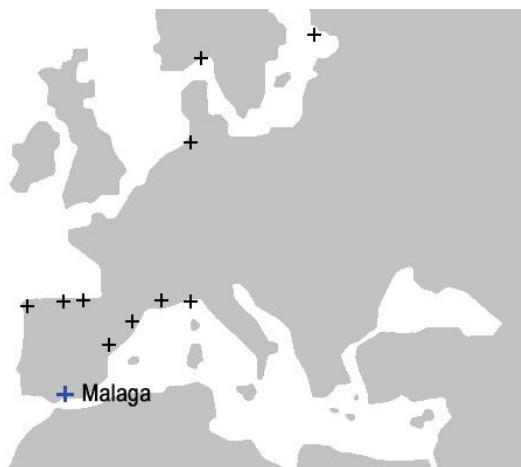

**Figure 1.** Malaga. Spain. UTM: N36°43'12.58" O4°25'13.22".

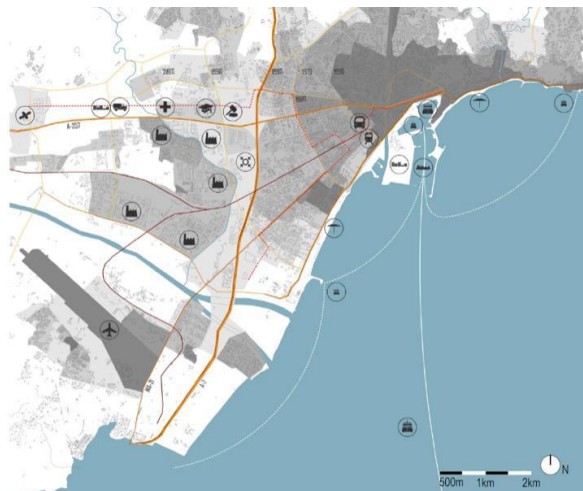

**Figure 2.** Main uses of city/port. Source: own elaboration.

**Table 4.** Values of indicators in the case of Malaga. 2012–2019.

| MALAGA | 2012 | | 2019 | |
|---|---|---|---|---|
| Indicator | Value | Rank | Value | Rank |
| 1.Physical Continuity (PDc) | $B = 2\ Pap = 0$ | 1 | $B = 2\ Pap = 1$ | 2 |
| 2.Continuity of Movement (PDm) | $PTm = 0\ PTt = 1$ | 2 | $PTm = 0\ PTt = 1$ | 2 |
| 3. Visual Continuity (PDv) | $Vp = 2\ Va = 1$ | 3 | $Vp = 3\ Va = 1$ | 4 |
| 4.Architectural Quality (PDq) | $Qp = 2\ Qb = 2\ Qps = 1$ | 3 | $Qp = 2\ Qb = 2\ Qps = 1$ | 3 |
| 5.Blending Uses (FDu) | $Ud = 0\ Uu = 1$ | 2 | $Ud = 1\ Uu = 1$ | 3 |
| 6.Balancing Free Spaces (FDps) | $Pss = 1\ Psd = 1\ Psb = 2$ | 5 | $Pss = 1\ Psd = 1\ Psb = 2$ | 5 |
| 7.Focal Uses Locations (FDfu) | $Fub = 3\ Fc = 0$ | 3 | $Fub = 3\ Fc = 0$ | 3 |
| 8.Flexibility in spaces and buildings, (FDfl) | $Fl = 10\text{-}25\ years\ Flr = 1\ Fld = 2$ | 4 | $Fl=10\text{-}25years\ Flr =1Fld=2$ | 4 |
| 9.Environmental Integration (SDe) | $Em = 1\ Emo = 2\ Ep = 0$ | 3 | $Em = 1\ Emo = 2\ Ep = 0$ | 3 |
| 10.Public Image and Support (SDps) | $Pp = 1\ Pa = 2\ Pws = 2$ | 3 | $Pp = 1\ Pa = 3\ Pws = 3$ | 5 |
| 11.Education and Employment (SDee) | $Ep = 0\ Et = 1\ Ei = 0$ | 2 | $Ep = 1\ Et = 1\ Ei = 1$ | 4 |
| 12.Integrating the Port with the City (SDpc): | $Pct = 0\ Pcd = 0\ Pcl = 0$ | 1 | $Pct = 0\ Pcd = 0\ Pcl = 1$ | 2 |

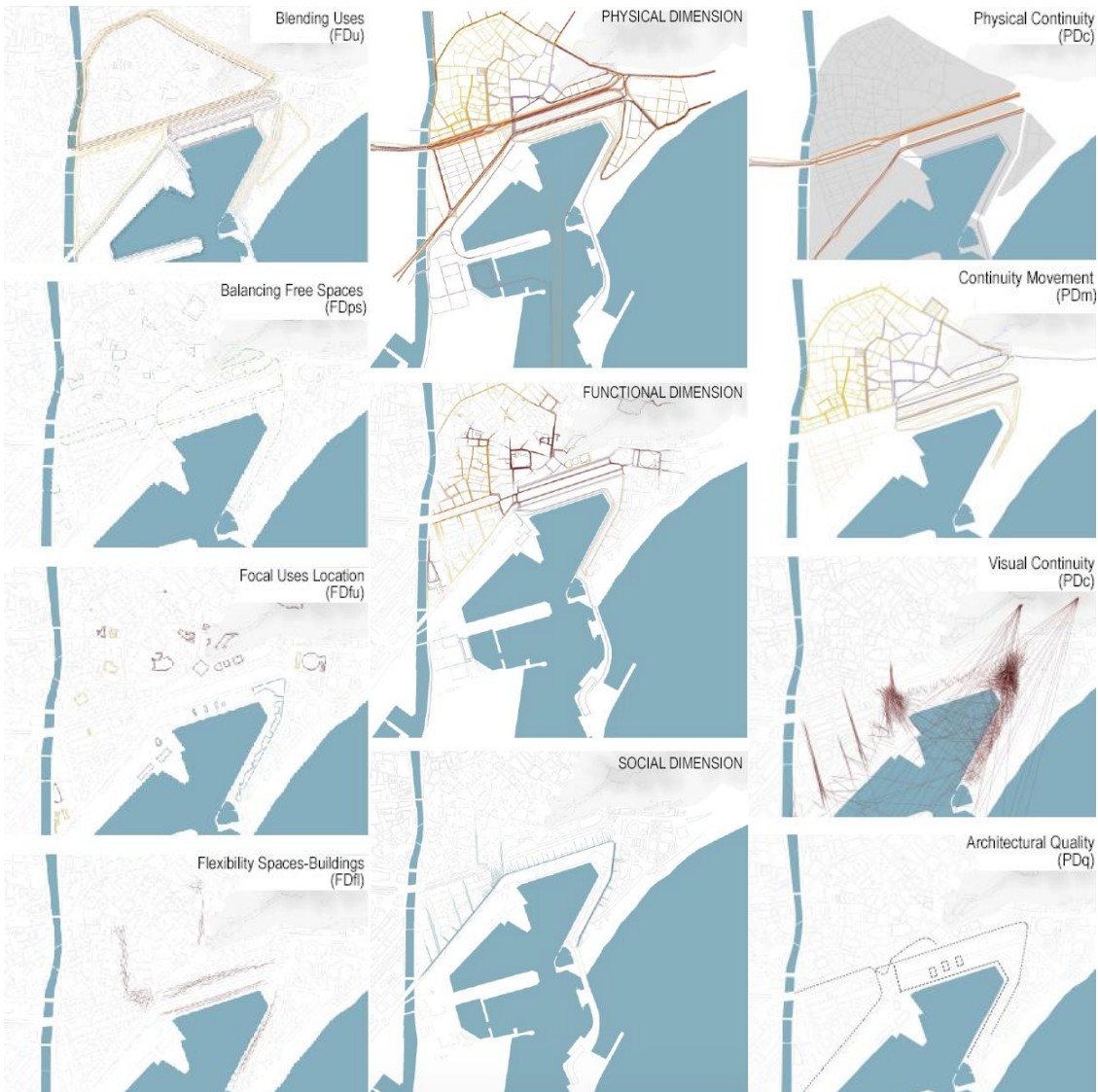

**Figure 3.** Graphic explanation of the analysis performed in Malaga. Source: own elaboration.

1    Physical Continuity (PDc):

*B:* In 2012, Malaga had a big barrier, a twelve-lane road, between the port and city, and in 2019 this remains;

*Pap*: In 2012, only two crosswalks crossed the barrier; in 2019 there are three.

The Physical Continuity has improved from PDc = 1 to PDc = 2, but it is still poor.

2    Continuity of Movement (PDm):

*PTm:* There was not maritime public transport in Malaga in 2012 and neither is there in 2019;

*PTt:* The terrestrial public transport reaches only the port–city border area in both years.

3    Visual Continuity (PDv):

*Vp:* In 2012, there were some views of the port from different streets but, in 2019, the view of the port from the city is greater and viewpoints have originated on the terraces of the hotels which further increase the presence of the port in the city;

*Va:* However, the atmosphere of the port has not yet reached the shops on either date.

4    Architectural Quality (PDq):

There has been no change in the architectural level in the waterfront, so the values are maintained on both dates.

*Qp:* The town planning is quite comprehensive and with a medium-term vision, and it has not changed from 2012 to 2019;

*Qb:* The heritage has not been preserved, but the new buildings have an architectural quality and generate an urban environment around them;

*Qps:* The public spaces have a high quality, and both citizens and tourists walk and live in these spaces.

5    Blending Uses (FDu):

*Ud:* From 2012 to 2019, port activity has increased in urban docks, with ferries and some berths;

*Uu:* Leisure uses mixed with cultural uses are maintained.

6    Balancing Free Spaces (FDps):

*Pss:* The public space is very wide and of great quality;

*Psd:* There is a low berth density;

*Psb:* A very low building density.

7    Focal Uses Locations (FDfu):

*Fub:* In the case of Malaga, no major international events have been held, so its buildings are small and medium sized and fit the scale of the city, focused on shops, restaurants and leisure;

*Fc:* In Malaga, the city centre and the port are less than 500 meters away, however, they have no uses that complement and generate synergies with each other.

8    Flexibility in spaces and buildings, (FDfl):

*Fl:* A 10–25 years lease is a medium term that hinders the adaptability of buildings and uses;

*Flr*: There are not revertible buildings in this waterfront;

*Fld:* However, despite the lack of flexibility, the density of buildings is low.

9    Environmental Integration (SDe):

*Em:* The environment's management is a very important issue to allow the permanence of an operative port close to the city centre of a city whose main industry is tourism. Although there is an environmental sustainability plan, this factor can and should continue to improve;

*Emo:* This port has a high level of environment monitoring;

*Ep:* Energy production is a pending issue in the case of Malaga, as it has the perfect conditions to do this, with 3.000 h of sunlight per year.

10    Public Image and Support (SDps):

*Pp:* This port has developed measures to understand public perception of the port and cruise activity;

*Pa:* The port has improved its strategies for continuous visitors and there is a port visitor information centre and a Sea Museum;

*Pws:* From 2012 to 2019, the port of Malaga has improved greatly with social networks.

11    Education and Employment (SDee):

*Ep:* There is no university degree related to the port, however there are relationships with research groups of the university, in economics, architecture and tourism, that work with the port;

*Et:* The training of port workers is encouraged;

*Ei:* The information on labour activity in the port has improved significantly, tripling school visits to the port.

12    Integrating the Port with the City (SDpc):

*Pct:* There are not mobility networks between the port and the city;

*Pcd:* There are also no daily uses in this waterfront beyond leisure uses such as restaurants and clothing stores and museums;

*Pcl:* In the seven years studied, the waterfront of Malaga has become a high-quality place of leisure, contemplation and enjoyment.

Although the port city of Malaga has improved the lower indicators, these remain insufficient. The great physical barrier causes a huge lack of continuity (PDc) and movement (PDm) that prevents the development of daily uses in the port (FDu), and therefore integrates the port in the day-to-day life of citizens (SDpc) (See Table 5).

The three interaction dimensions were analysed for each case as being complementary and dependent, i.e., one cannot work without the others. However, the intensity and apportionment of the indicators in each dimension are key to success. The proportions depend on the local context, the reality of the port and its inhabitants. Even with the criteria proposed and proven in these twelve cases, the correct prescription for a port city can only be arrived at through a good diagnosis of the problems at that location (See Figure 4).

According to the analysis carried out, the success of the waterfront transformations in these case studies is directly proportional to the quality of the planning. Most show a global vision that is implemented in a concerted manner with the support of the port authority, city council and local population. Key directives are drawn from the overall concept to establish guidelines for subsequent local implementation.

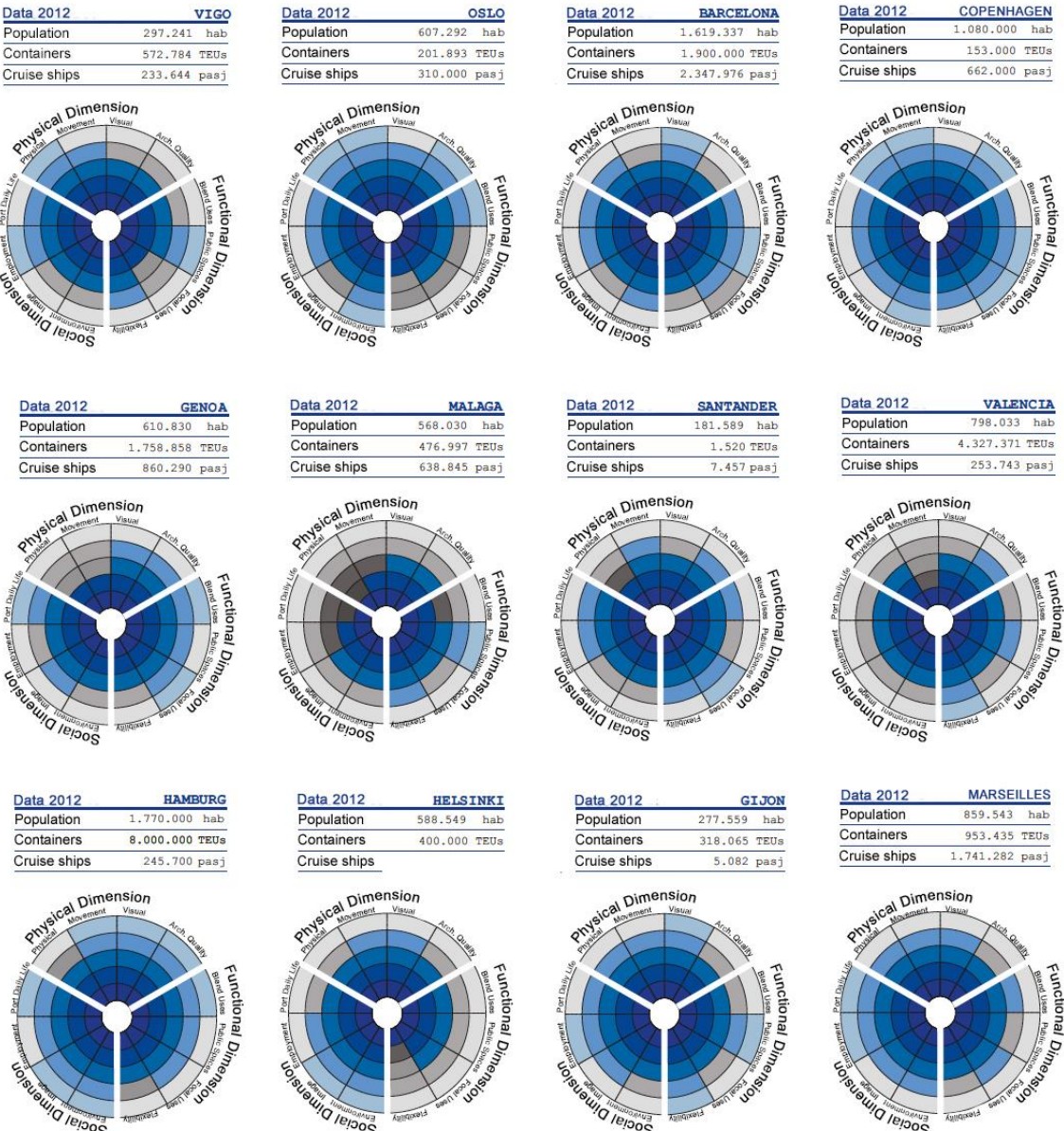

**Figure 4.** Applying 3DPortCityMeasure: comparative visual analysis of port–city integration.

**Table 5.** Values of indicators in the Cases of Study. 2012.

| Case Collection | P (inhab) | Cruisers | Containers | PDc | PDm | PDv | PDq | FDu | FDps | FDfu | FDfl | SDe | SDps | SDee | SDpc |
|---|---|---|---|---|---|---|---|---|---|---|---|---|---|---|---|
| Vigo (SP) | 297,241 | 233,644 | 572,784 | 5 | 4 | 3 | 3 | 3 | 5 | 2 | 4 | 3 | 3 | 5 | 4 |
| Oslo (NW) | 607,292 | 310,000 | 201,893 | 5 | 5 | 4 | 5 | 5 | 3 | 3 | 2 | 5 | 3 | 4 | 4 |
| Barcelona (SP) | 1,619,337 | 2,347,976 | 1,900,000 | 4 | 3 | 5 | 3 | 4 | 5 | 4 | 3 | 4 | 3 | 4 | 4 |
| Copenhagen (DK) | 1,080,000 | 662,000 | 153,000 | 5 | 5 | 4 | 5 | 4 | 5 | 5 | 4 | 5 | 4 | 4 | 4 |
| Genoa (IT) | 610,830 | 860,290 | 1,758,858 | 2 | 2 | 4 | 4 | 5 | 4 | 5 | 3 | 3 | 4 | 3 | 5 |
| Malaga (SP) | 568,030 | 638,845 | 476,997 | 1 | 2 | 3 | 3 | 2 | 5 | 3 | 4 | 3 | 3 | 2 | 1 |
| Santander (SP) | 181,589 | 7457 | 1520 | 2 | 4 | 3 | 4 | 4 | 3 | 5 | 4 | 3 | 3 | 4 | 4 |
| Valencia (SP) | 798,033 | 253,743 | 4,327,371 | 3 | 1 | 3 | 4 | 3 | 4 | 3 | 5 | 3 | 3 | 3 | 3 |
| Hamburg (DE) | 1,770,000 | 245,700 | 8,000,000 | 3 | 5 | 5 | 5 | 5 | 4 | 4 | 3 | 5 | 5 | 4 | 5 |
| Helsinki (FI) | 588,549 | 360,000 | 400,000 | 3 | 4 | 4 | 3 | 3 | 3 | 3 | 1 | 5 | 5 | 4 | 3 |
| Gijon (SP) | 277,559 | 5082 | 318,065 | 4 | 4 | 5 | 3 | 3 | 5 | 3 | 5 | 3 | 4 | 5 | 4 |
| Marseille (FR) | 859,543 | 1,741,282 | 953,435 | 4 | 4 | 4 | 3 | 4 | 3 | 4 | 3 | 4 | 4 | 5 | 5 |

### 4.1. Physical Dimension (PD)

The study of the different interventions shows the importance of physical continuity in the processes of port–city integration. It is insufficient when pedestrian access is provided only occasionally. Several crossings are needed along the length of the sea front, and it is recommendable to extend the city's main pedestrian streets through the docks. Cities like Vigo, Oslo, and Copenhagen have total pedestrian continuity between the city and the port that empowers and greatly facilitates the relationship between both realities. In other cases, such as Malaga, twelve lanes of road traffic separate the port from the city, which makes the connection very difficult, even if they invest in urban uses and quality spaces.

One of the most influential guidelines for port–city integration is for the urban mobility plan to include a maritime public transport network to complement the terrestrial one. As is demonstrated in cities such as Hamburg, Oslo, and Copenhagen, water-borne transport is not only one of the most sustainable forms, but it also integrates the port and the sea into the population's everyday activities, and thus becomes part of their daily lives. Boarding points for this maritime transport have great potential, both in their primary function and combined with other uses such as landscape enhancement, as they serve as vantage points for observing marine activity in the port.

### 4.2. Functional Dimension (FD)

A widely used tactic has been to host international events to fund a large part of waterfront interventions. Most of these opportunities are used to improve the city's infrastructure and public image. However, in many cases, this has resulted in large buildings to accommodate temporary events. Although they are often planned to be multipurpose for subsequent adaptation to some as yet undecided use, they still end up leaving the port as an architectural theme park. The most successful cases are those that have established a balance between these large buildings and other smaller ones that respond to the inhabitants' own needs, preserving as much of their heritage features as possible to maintain the memory of the place. When urban uses are introduced, the docks interact with a city's historic centre, so that the city's uses and the urban ones provided by the port complement each other, generating synergies between both realities. That is the case in Genoa, where students continually cross from the city to the port because the university is on both sides, along with the fruit and vegetable market in front of the fish market in the port, and museum routes that go from the port to the city. Oslo, Copenhagen and Hamburg are also cities in which daily urban uses are located in their ports. In other cases, the uses of the port are extremely zoned, such as shopping areas or residential areas, and even cultural zones. Zoning creates a very specific audience within a certain time slot.

### 4.3. Social Dimension (SD)

It is increasingly important to win public support when transforming the port front. In fact, the most recent interventions began the process even before the planning stage, like Marseille, to improve its image and encourage the inhabitants to support their port. To achieve this, it is essential to promote education and outreach programs through different channels to encourage awareness among the population about the port, its history, its activity and its impact on the city, just as Gijon does. The more people get to know the port, the more they will become involved with it, and the more they will appreciate it and admire it and want it to be part of their daily lives. Providing vantage points to watch the port at work, as happens in Hamburg, throughout Landungsbrücken; organizing events in the docks as in Helsinki; preserving tangible and intangible heritage and publicizing the port's own culture, such as storytelling, gastronomy, language, photographs, and anecdotes, are some of the initiatives that are carried out to enhance the port's identity in the city.

## 5. Discussion

Throughout the history of waterfronts, since the first American experiences in the 1960s and 1970s, there have been hundreds of cities that have transformed their historic port with urban uses, maintaining, to a greater or lesser degree, port activities. These interventions went from city to city, with increasingly banalized models and without really studying the context of each revitalization project. Subsequently, studies and research into specific cases began to emerge and, in some cases, up to two and three cities were studied, but there is no methodology to study and compare these interventions.

Between 2007 and 2010, three different guides were published that refer to good practices in waterfront interventions, exposing cases of specific cities.

The first of these guides is *The Cool Sea. Waterfront Communities Project Toolkits* [38]. It is a very interesting document that collects a research project carried out by nine cities of the North Sea, and their ports and universities. The project recognizes their potential contribution to the sustainable movement of people and freight, including an increase in short sea shipping. The guide exposes 11 quite generic actions, and each one is carried out by one of the nine cities over a limited scope, such as the North Sea.

The second guide is *Plan the City with the Port, guide of good practices* [40] realized by AIVP—International Association Cities and Ports. The purpose of this guide is to provide a decision support tool for addressing the problems that will be faced when putting into practice the ideal of planning the city's integration with the port. The guidelines contained herein, and the examples provided, do not claim to be exhaustive, just sources of inspiration to address four major topics: spatial organization; economic development strategies; environmental challenges; and project management and governance.

The third one is the *Code of Practice on Societal Integration of Ports* [39] carried out by ESPO—European Sea Ports Organisation. It is a very complete guide of good practices for ports focused on social integration, where more concrete actions are exposed than in the first guide and within a larger scope, since it covers examples from all over Europe.

None of these guides present a comparative study or an attempt to quantify port–city integration. They offer a partial and biased view of the complexity of the waterfront phenomenon.

With 3DPortCityMeasure, we propose a methodology based on three interaction dimensions (physical, functional and social interaction) and a series of indicators for each, which enables us to measure the level and therefore the quality of port–city integration processes.

The analysis of the twelve proposed cases confirms, on the one hand, the complexity of the interactions between these two port–city realities that are so difficult to measure, and on the other, that the results are satisfactory. These results demonstrate the usefulness of the tool, which has been developed by enabling the production of comparative graphs to visualize this extremely complex reality.

The research provides agreed indicators that allow us to empirically contrast the strategies developed for the port–city relationship in each case, including the subjective factors characteristic of the port–city relationship. 3DPortCityMeasure is a structured, quantitative and qualitative method based on multiple criteria of experts' judgement to obtain the priorities of several alternatives.

The results show that by characterizing these urban waterfront systems using the three proposed interaction dimensions, the highly complex port–city relationship can be more easily understood. This framework is also applicable at different geographic scales, and even to other urban studies that analyse interactions of different natures. However, the use of the tool in other cases and the continuity of its use at different periods of the same case would contribute to validate its usefulness.

Furthermore, it is important to mention that some aspects of it that could be improved. The first is characterized by the current surge in cruise activity, that in several cases is unbalancing the functional interaction between the port and the city [55]. The second should be considered in the social dimension, as it refers to the relationship between the planning and management of spaces and buildings resulting from waterfront interventions, and its economic impact on the city [56].

## 6. Conclusions

In recent years, the integration of cities with their old port spaces has been based on, principally, the good practices built up over more than fifty years, which represent a considerable body of knowledge and experience.

Contemporary perspectives on urban renewal are associated with an orientation towards sustainability, resilience or social and territorial cohesion, and these in turn determine perspectives on port–city integration. These involve not only repurposing the working port's productive and industrial structures, and integrating the historic part with the city, but also promoting economic activity and employment with actions that are innovative and environmentally and energy efficient.

In this way, the main finding of the present research is the ability of the proposed methodology to express, in a simple visual diagram, the complex port–city relationship.

The utility of collecting successful experiences of port–city integration and extracting a series of indicators that allow the quantification of different port–city cases integration was evidenced.

In this process, the literature review was used as a valuable complementary method to the expert knowledge to define the methodology and completed with the databases of each city–port–university for the assignment of different values.

3DPortCityMeasure allows a comparative study of the different port cities to be carried out with common indicators, enabling it to learn and improve with the transfer of knowledge and experiences between different cases.

According to the analysis carried out, the success of the waterfront transformations in these case studies is directly proportional to the quality of the planning. Most show a global vision that is implemented in a concerted manner with the support of the port authority, city council and local population (SD) for a long-term process. The study of the different interventions shows the importance of physical continuity in the process of port–city integration (PD), such as extending the city´s main pedestrian streets through the docks, as well as a maritime public transport network that complements the terrestrial one. The most successful cases establish a balance between port and urban uses (FD), with the latter arranged both in the city and in the port, generating synergies between them.

The intervention framework reaches beyond the immediate planning area. Regardless of its size, this area must also take into consideration the maritime–terrestrial edge space, the points of contact between port and city, and other border areas affected by the mutual interactions between urban and port uses. It is difficult to focus the resolution of a new urban planning project without analysing the border-line space as a whole. It is precisely an understanding of the space as a whole, even at the city-wide level, of the situations that occur in it, and of the needs and requirements of port activities and urban uses, that gives rise to the specific measures and concrete decisions needed for planning and for projects.

These projects are a key opportunity to promote sustainable economic and social development and should not be wasted by short-term thinking or exclusively commercial interests. To quote Dirk Schubert, 'This is not some passing fad of postmodern urban development, but a unique historical possibility for urban waterfront development and port–city reintegration' [34].

The planning processes require a broad vision and a highly coherent overall spatial framework to reap the maximum benefits of development and regeneration. That vision must be clear and widely accepted, and it must be specific regarding the objectives for urban life in the scope of action. Regeneration cannot be achieved with classic, formal planning as it is not suited to the greater complexity of today's cities. A wide range of strategies is needed, and they have to be constantly updated to adapt to this dynamic and changing context.

It is critical to keep in mind that this is an historic opportunity that has to be carried out in the long term, and 3DPortCityMeasure provides a clear and simple way to continuously evaluate the process and the decision-making, and to adapt strategies to changing contexts through a consolidated knowledge base.

3DPortCityMeasure is a methodology motivated by the need for a common framework for analysing complex port–city integration from a society perspective. This can be considered as a preliminary work, the basis of further research, in order to find a more exact methodology.

**Author Contributions:** Conceptualization, M.J.A. and J.B.L.; Methodology, M.J.A. and J.B.L.; Validation, M.J.A., J.B.L. and J.P.C.; Formal Analysis, M.J.A.; Investigation, M.J.A. and J.B.L.; Data Curation, M.J.A. and J.B.L.; Writing–Original Draft Preparation, M.J.A. and J.P.C.; Writing–Review & Editing, M.J.A. and J.P.C.; Visualization, M.J.A. and J.P.C. All authors have read and agreed to the published version of the manuscript.

**Funding:** This research received external funding from Universidad de Malaga.

**Acknowledgments:** The authors are grateful to the editor and the anonymous reviewers, whose helpful remarks on the previous drafts provided important inputs for this paper's final form.

**Conflicts of Interest:** The authors declared no potential conflicts of interest with respect to the research, authorship, and/or publication of this article.

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
