# Peer review of "3DPortCityMeasure: Methodology for the Comparative Study of Good Practices in Port–City Integration"

_sustainability, doi:10.3390/su12030880_

Round 1

Reviewer 1 Report

The 4 indicators for the Physical Dimension seems all descriptive and qualitative. Likewise for the Functional Dimension and Social Dimension.

Each of the 12 indicators can reach a rank from 5 to 1: 5 in the worst definition case, 1 in the better definition case. How are associated the scores to the individual cases?

In § 4 “Applying the Framework”:

“The complete process followed and the details of the investigation are available in Andrade et al. (2012).” I think that can be necessary a more wide description of the process. Also to explain better the Graphs of Fig. 1

After the Fig. 1 there is a description of the results for the three Dimensions. I think it is necessary to deep these considerations with the aim to make stronger the results applyed to the nuances of the specific dynamic of each city.

Author Response

The authors are grateful to the reviewers for their helpful suggestions to improve this paper. Following their comments, the paper has been updated from its previous version.

The paper has been deeply reviewed. Each section has been improved, with special attention to the explanation of the methodology, including a new table, and explaining and defining each indicator and its value, as well as the results in section 4.

A new Discussion section has been created explaining the findings of the methodology based on previous studies as well as its limitations.

The revisions have been clearly highlighted in blue text. It was with the "Track Changes" marks function in Microsoft Word, but when the document was sent for editing review, all the marks of the review disappeared. That is why we have chosen to mark the text modified in blue.

The answer to each comment is provided below.

The 4 indicators for the Physical Dimension seems all descriptive and qualitative. Likewise for the Functional Dimension and Social Dimension.Each of the 12 indicators can reach a rank from 5 to 1: 5 in the worst definition case, 1 in the better definition case. How are associated the scores to the individual cases?

Explained pages 7-12.

In § 4 “Applying the Framework”:“The complete process followed and the details of the investigation are available in Andrade et al. (2012).” I think that can be necessary a more wide description of the process. Also to explain better the Graphs of Fig. 1

Explained pages 13-15

After the Fig. 1 there is a description of the results for the three Dimensions. I think it is necessary to deep these considerations with the aim to make stronger the results applyed to the nuances of the specific dynamic of each city.
The explanation of some data has been extended. The totality has not been done so as not to extend the document further

Reviewer 2 Report

It is an interesting and valuable work which explores not too obvious associations between port and city. 

The importance of integration / continuity of ports into the city fabric seems not sufficiently justified in the introduction.There are different types of ports and not all of them seem safe, interesting places to be integrated into urban fabric. To this end authors need to be clearer of their vision (for example, is the goal to connect the city to industrial ports which currently are closed to the public? should the word "revitalization" fit to describe authors approach?). 

Content from the page 2 and 4 (about 4 lines of text) is repeated.

Stylistically the manuscript needs improvements , shorter and more concise sentences, removing redundancy, which could make the leading thought unclear.

There are some rather confusing sentences, for example: "Each of these dimensions, these three blocks, derives chronologically from the different stages of waterfront evolution and consequently indicates the period that came after .."; "While physical integration is not the most important in the port-city relationship, it is fundamental to it."; "In this last point we are referring to maintaining the identity of the port spaces: Not turning the dockside into a city, but rather adapting it to urban life, to enable city inhabitants to meet there but without losing the port atmosphere, the memory of the place." and more... Stylistic check by a native English speaker is necessary.

On page 4, authors start sentences from "Secondly..", "Third.." but there is no "first", this is confusing and could be corrected.

What methods are used to evaluate different complex aspects of the port integration - for example, where the data on the pollution caused by port (indicator 9) is derived from?

The selection of case studies demonstrates the feasibility of developped framework in European port-cities. Hence, it is speculative to define the framework as applicable globally as authors state. 

Figure 1 not readable.

Is the developped framweork an expert-based tool? Also, it should be reported who performed evaluation of 12 case-studies

The mere fact authors used the developped tool is not yet a proof that the tool is useful and applicable in many contexts. To do so a validation or reliability study should be performed. 

Author Response

The authors are grateful to the reviewers for their helpful suggestions to improve this paper. Following their comments, the paper has been updated from its previous version.

The paper has been deeply reviewed. Each section has been improved, with special attention to the explanation of the methodology, including a new table, and explaining and defining each indicator and its value, as well as the results in section 4.

A new Discussion section has been created explaining the findings of the methodology based on previous studies as well as its limitations.

The revisions have been clearly highlighted in blue text. It was with the "Track Changes" marks function in Microsoft Word, but when the document was sent for editing review, all the marks of the review disappeared. That is why we have chosen to mark the text modified in blue.

The answer to each comment is provided below.

It is an interesting and valuable work which explores not too obvious associations between port and city. 

The importance of integration / continuity of ports into the city fabric seems not sufficiently justified in the introduction. There are different types of ports and not all of them seem safe, interesting places to be integrated into urban fabric. To this end authors need to be clearer of their vision (for example, is the goal to connect the city to industrial ports which currently are closed to the public? should the word "revitalization" fit to describe authors approach?). 

Yes, we agree. The first paragraph of Section 2 has been moved to the introduction so that it is understood that the relationship is with those docks that remain after the new location of the industrial port. And a new explanatory paragraph has been introduced:“In many cases, the transformation of disused docks has changed the course of the city’s future development, but there have also been failures. Therefore, it is essential to learn from the strategies and actions that have improved the quality of life of the inhabitants by increasing the economy and local well-being, compared to many others that have not been so successful".

Content from the page 2 and 4 (about 4 lines of text) is repeated.

Solved

Stylistically the manuscript needs improvements, shorter and more concise sentences, removing redundancy, which could make the leading thought unclear.

The article has been sent and reviewed by the MDPI English Editing Service

On page 4, authors start sentences from "Secondly..", "Third.." but there is no "first", this is confusing and could be corrected.

Solved

What methods are used to evaluate different complex aspects of the port integration - for example, where the data on the pollution caused by port (indicator 9) is derived from?
More information on the indicators has been added, as well as a table indicating the websites of each city and port where the data is taken and updated

The selection of case studies demonstrates the feasibility of developped framework in European port-cities. Hence, it is speculative to define the framework as applicable globally as authors state. 
It is true that the study is carried out for European cities, although it has also been applied to cities like Sydney. A Discussion section is added explaining the limitations and future research of the methodology.

Figure 1 not readable.
The figure is attached to more resolution

Is the developped framweork an expert-based tool? Also, it should be reported who performed evaluation of 12 case-studies.
The study was conducted by three experts in the field

The mere fact authors used the developped tool is not yet a proof that the tool is useful and applicable in many contexts. To do so a validation or reliability study should be performed. 

A discussion section is added explaining the limitations of the current tool as well as the possibilities of it with future research.

Reviewer 3 Report

The subject may be worthy of research, however the article has serious shortcomings in its scientific approach. Both the content and the structure of the article require significant improvements from the point of view of scientific soundness. In this sense, aspects such as the review of the state of the art, the methodological approach or the presentation and scientific discussion of the results are quite poor. Even so, I think there is room for improvement if the authors are willing to do the work necessary to incorporate substantial modifications. Accordingly, I recommend a major revision indicating to the authors that currently my opinion is closer to a rejection or resubmission than to an acceptance. Below I detail the main issues that should be addressed by the authors: 

LITERATURE REVIEW 

Among 46 references only 3 correspond to research articles from relevant scientific journals. The rest of references quotes architecture/urban planning books, local (no international) publications, no relevant research journals (or at least not indexed with impact factor), etc. In addition, the bibliography is in general quite obsolete (no recents research publications, being the majority of references from the 90s). By last, a comprehensive review of the state of the art is missing from a methodological point of view (authors here only cite for that books from architecture or urban planning regarding the topic). All of this is not admisible for an article proposed for consideration in a international research journal such as Sustainability.

METHODOLOGY 

The significance of the content and its scientific soundness are quite questionable from a methodological point of view. Authors are proposing a new framework to assess?analize?diagnose? port-city integration. This supposed innovative proposal is applied to 12 case studies with 4 indicators valued from 1 to 5 for 3 different fields. Nevertheless, these values seem to be assigned in a subjective way by authors since no numerical or spatial analysis with formulas is proposed by them (or at least they result from a “black box” to the readers of the journals). Should we understand that the analysis remain simply qualitative? In a summary, the main contribution of the article is only the proposal of 12 qualitative indicators to analyze port-city integration? 

By the other hand, these 3x4=12 values allow authors to present 12 different figures, one for each case study. What is the consequence for policy implications to apply in those cities? How does this framework can help decision makers or urban planners of cities?

In my opinion some kind of more relevant and concrete numerical approach would be necessary to make the framework reliable, scientifically robust and helpful to stakeholders that may be interested in this topic. See good similar proposals of  frameworks applied to different city case studies using robust, objective and justified scientific approaches such as for example:

Strategic urban planning in cities through AHP approach for decision support processes in Campos-Sánchez, F.S.; Reinoso-Bellido, R.; Abarca-Álvarez, F.J. Sustainable Environmental Strategies for Shrinking Cities Based on Processing Successful Case Studies Facing Decline Using a Decision-Support System. Int. J. Environ. Res. Public Health, 2019, 16, 3727.

Territorial anthropization evaluation in coastal areas through GIS indicators and stakeholders panel decision in García-Ayllón, S. GIS Assessment of Mass Tourism Anthropization in Sensitive Coastal Environments: Application to a Case Study in the Mar Menor Area. Sustainability 2018, 10, 1344.

STRUCTURE

The article lacks of a real discussion section. Results are presented and then the authors introduce some kind of strange conclusion section where personal opinions, citations and a summary of the manuscript are mixed, but a relevant scientific discussion is missing. On the one hand, authors should incorporate a clear specific discussion section to compare and show improvements and differences/similarities with previous studies, comment the limitations of the new methodology, analyze policy implications, suggest future lines of research, etc. (see journal template for the content of this section). On the other hand, a clear conclusion section must be done to only summarize the main findings of the article so that the readers of the journal can assess their interest in the article at a glance.

OTHER MINOR ISSUES
- Use MDPI standards for references, bibliography, typography, etc.

Author Response

The authors are grateful to the reviewers for their helpful suggestions to improve this paper. Following their comments, the paper has been updated from its previous version.

The paper has been deeply reviewed. Each section has been improved, with special attention to the explanation of the methodology, including a new table, and explaining and defining each indicator and its value, as well as the results in section 4.

A new Discussion section has been created explaining the findings of the methodology based on previous studies as well as its limitations.

The revisions have been clearly highlighted in blue text. It was with the "Track Changes" marks function in Microsoft Word, but when the document was sent for editing review, all the marks of the review disappeared. That is why we have chosen to mark the text modified in blue.

The answer to each comment is provided below.

Among 46 references only 3 correspond to research articles from relevant scientific journals. The rest of references quotes architecture/urban planning books, local (no international) publications, no relevant research journals (or at least not indexed with impact factor), etc. In addition, the bibliography is in general quite obsolete (no recents research publications, being the majority of references from the 90s). By last, a comprehensive review of the state of the art is missing from a methodological point of view (authors here only cite for that books from architecture or urban planning regarding the topic). All of this is not admissible for an article proposed for consideration in an international research journal such as Sustainability.

In the field of study of the Waterfront and the port-city relationship, there is a greater tendency to books than journal articles. These journals are specialized in the subject and although they are not indexed but they are in multiple databases such as Avery, Riba.... The books are written by great experts and are a benchmark for researchers. The research presented is based on the history of waterfronts, hence there is a lot of bibliography from the 90s, and the database sources are websites of ports and cities. The bibliography presented is limited exclusively to that cited in the article. Although they are not cited in the text, multiple articles on methodology have been studied. However, bibliography has been expanded in current articles in scientific journals focused on methodological study.

The significance of the content and its scientific soundness are quite questionable from a methodological point of view. Authors are proposing a new framework to assess?analize?diagnose? port-city integration. This supposed innovative proposal is applied to 12 case studies with 4 indicators valued from 1 to 5 for 3 different fields. Nevertheless, these values seem to be assigned in a subjective way by authors since no numerical or spatial analysis with formulas is proposed by them (or at least they result from a “black box” to the readers of the journals). Should we understand that the analysis remain simply qualitative? In a summary, the main contribution of the article is only the proposal of 12 qualitative indicators to analyze port-city integration? 

The proposal is innovative because it allows quantifying intangible values ​​and thus being able to carry out a comparative study of different port-cities. Until now, there is no study that analyses a series of common factors in several waterfront cases. In turn, these factors are grouped to simplify the complexity of the port city relationship. Table 2 shows what the assignment of values ​​from 1 to 5 corresponds to. This assignment is not subjective, however, there is no specific mathematical formula. A further explanation of the methodology is added to the text. Undoubtedly, the comments are very interesting and in a future investigation it would be a great challenge to be able to develop mathematical formulas for each of the factors presented as well as expand the number of factors as explained in the added section of Discussion.

By the other hand, these 3x4=12 values allow authors to present 12 different figures, one for each case study. What is the consequence for policy implications to apply in those cities? How does this framework can help decision makers or urban planners of cities?

In my opinion some kind of more relevant and concrete numerical approach would be necessary to make the framework reliable, scientifically robust and helpful to stakeholders that may be interested in this topic. See good similar proposals of  frameworks applied to different city case studies using robust, objective and justified scientific approaches such as for example:...

The proposed literature is appreciated, which is of great interest. The utility for decision makers and urban planners is both the matrix that allows to compare the city with other success cases, as well as the evolution of the city over time. The factors that are applied to quantify the port-city relationship are based on success cases, being recommendations not only in one part, but in the physical, functional and social dimension, so that it offers a balance in the evolution of the strategy proposals of each city.

A review of the methodology section of the paper is carried out providing a more numerical approach and considering the recommended literature.

The article lacks of a real discussion section. Results are presented and then the authors introduce some kind of strange conclusion section where personal opinions, citations and a summary of the manuscript are mixed, but a relevant scientific discussion is missing. On the one hand, authors should incorporate a clear specific discussion section to compare and show improvements and differences/similarities with previous studies, comment the limitations of the new methodology, analyze policy implications, suggest future lines of research, etc. (see journal template for the content of this section). On the other hand, a clear conclusion section must be done to only summarize the main findings of the article so that the readers of the journal can assess their interest in the article at a glance.

 A Discussion section has been added and the Conclusion section revised

OTHER MINOR ISSUES
- Use MDPI standards for references, bibliography, typography, etc.

Solved

Thank you very much

Round 2

Reviewer 2 Report

The manuscript has undergone a major revision, and visible improvements. All comments and suggestions have been clearly addressed and resolved, so the manuscript is much more comprehensive. Congratulations to the team for their great work.

One remaining issue is with the English revision. In the current version of the manuscript seemingly inapropriately composed sentences are still apparent, (for example: "These cases are at very different scales, with different features, uses, morphologies, situations and locations in the city. However, they have found great social acceptance and have greatly improved the urban quality of their cities, are examples of sustainability, and have maintained the spirit of the port in the waterfront."). It is suggested that authors double check their versions. 

Author Response

Thank the reviewer for their suggestions. They have been carried out.

Reviewer 3 Report

The authors have improved the content of the manuscript on some issues. However, the main problems of the manuscript highlighted in my previous report still present. In my opinion the global problem is that the manuscript looks more like a chapter in an architecture or urbanism book, than a research article from a relevant international scientific journal such as Sustainability. Consequently, I must reiterate the need for an in-depth review to enhance the research approach of the text, noting the authors that in case of not being attended to, my recommendation will possibly be to discourage the article for publication. Then I proceed to detail the issues that I understand that remain unresolved: 

METHODOLOGICAL RELEVANCE & SCIENTIFIC SOUNDNESS 

The scientific proposal is in my opinion quite rudimentary at the methodological level. We must understand then that the innovative contribution of this research to the field of investigation is simply the proposal of 12 qualitative indicators that the authors propose to value from 1 to 5? 

The results obtained in figure 1 (which are finally the only results presented in the article) at the end give the sensation to reader of coming from a “black box”. How have they been obtained by the authors? Each value from 1 to 5 has been obtained from: a specific qualitative assessment of each criteria stated for each city made by the author? a personal opinion of the author? or responds to a specific methodology of a technical nature? In the text, the only reference to how values ​​from 1 to 5 have been obtained is limited to saying that the detailed obtaining of the values ​​can be found in a study (local and therefore inaccessible to readers) done by the authors in 2012 (by the way, if the results were already published then what justifies now a new publication 8 years later?).

On the other hand, the importance of these 12 issues in the subject of port-city integration was already known (as state in the abundant bibliography existing) before the authors' proposal. Therefore, should it be understood that the scientific contribution corresponds to the parameterization of these 12 elements in five levels? Do these 12 issues have always a similar level of relative importance? Does its relative importance vary depending on the cities? Two good and relevant examples were proposed in my past report to ponder on how to improve the robustness and numerical approach of the framework to a more scientific one through.

LITERATURE REVIEW

The review of the art has a narrative rather than a research scientific approach. Authors almost do not include in the literature review relevant recent research papers from international scientific journals. The bibliography is still mainly composed by not recent urban planning and architecture books that address the issue from a narrative approach, and publications from local research or not indexed with impact factor journals. The justification for this from authors is that “In the field of study of the Waterfront and the port-city relationship, there is a greater tendency to books than journal articles”. This statement is quite questionable, even more so if it refers to the generation of a research background, as there are numerous scientific publications in relevant research journals made in recent years, such as:

Jianke Guo, Yafeng Qin, Xiaofei Du, Zenglin Han. Dynamic measurements and mechanisms of coastal port–city relationships based on the DCI model: Empirical evidence from China. Cities 96, 2020,102440

Jason Monios, Rickard Bergqvist, Johan Woxenius. Port-centric cities: The role of freight distribution in defining the port-city relationship. Journal of Transport Geography 66, 2018, 53-64.

C.A. Schipper, H. Vreugdenhil, M.P.C. de Jong. A sustainability assessment of ports and port-city plans: Comparing ambitions with achievements. Transportation Research Part D: Transport and Environment 57, 2017, 84-111.

Among others. The authors should eliminate narrative old architectural literature (even if their authors are relevant or well known urban planners, since Sustainability is not a architecture journal) and introduce research one to enhance the research approach of this section and generate scientific review of the state of the art rather than a retrospective narrative closer to a architecture essay. 

SIGNIFICANCE OF THE CONTENT

How would the stakeholders or decision makers apply the proposed framework in the urban planning of port and city? Is it just a tool aimed at urban diagnosis, or does it imply proposals for practical application? In this context, the usefulness of the proposed tool is not clear. What does the results of figure one mean? Probably the practical application in detail of the proposed process as a specific case study to one of the cities proposed would be advisable so that the reader of the journal can observe on one side how the values ​​indicated by the authors are obtained and on the other hand to know which practical implications the results have in the context of the policy implications.

By the way, the selection of cities respond to a specific criterion? Why are they all European? Would the framework be applied equally in Chinese or American cities?

Author Response

First of all thank you for your suggestions that have been very useful. The article has been thoroughly reviewed and a great effort has been made to explain the methodology. The literature review has been reviewed following the reviewer's advice.

METHODOLOGICAL RELEVANCE & SCIENTIFIC SOUNDNESS 

The results obtained in figure 1 (which are finally the only results presented in the article) at the end give the sensation to reader of coming from a “black box”. How have they been obtained by the authors? Each value from 1 to 5 has been obtained from: a specific qualitative assessment of each criteria stated for each city made by the author? a personal opinion of the author? or responds to a specific methodology of a technical nature? In the text, the only reference to how values ​​from 1 to 5 have been obtained is limited to saying that the detailed obtaining of the values ​​can be found in a study (local and therefore inaccessible to readers) done by the authors in 2012 (by the way, if the results were already published then what justifies now a new publication 8 years later?).

Obtaining the values ​​is explained and detailed with the application in a specific case.
The current publication focuses on the methodology. It is applied in the specific case on two different dates to demonstrate the usefulness of the tool as a self-control system.

On the other hand, the importance of these 12 issues in the subject of port-city integration was already known (as state in the abundant bibliography existing) before the authors' proposal. Therefore, should it be understood that the scientific contribution corresponds to the parameterization of these 12 elements in five levels? Do these 12 issues have always a similar level of relative importance? Does its relative importance vary depending on the cities? Two good and relevant examples were proposed in my past report to ponder on how to improve the robustness and numerical approach of the framework to a more scientific one through.

There is no study that presents these 12 factors. Section 2 explains how most studies focus on the economic development of the port, pollution... but not from a societal perspective, from the human factor. The relative importance of the 12 factors vary depending on the cities because of the diversity of port-city issues, but the ultimate goal, the optimal is to reach the highest score in the 12 factors as they are complementary.

LITERATURE REVIEW

The review of the art has a narrative rather than a research scientific approach. Authors almost do not include in the literature review relevant recent research papers from international scientific journals. The bibliography is still mainly composed by not recent urban planning and architecture books that address the issue from a narrative approach, and publications from local research or not indexed with impact factor journals.

The “2. Background” section has been changed following the reviewer's suggestions and incorporating more recent research and papers. Much of the narrative has been removed, maintaining only that which is subsequently related to the basis of the methodology.

SIGNIFICANCE OF THE CONTENT

How would the stakeholders or decision makers apply the proposed framework in the urban planning of port and city? Is it just a tool aimed at urban diagnosis, or does it imply proposals for practical application? In this context, the usefulness of the proposed tool is not clear. What does the results of figure one mean?

With the study of the case of Malaga in two different years, the usefulness of the tool in decision making can be better understood. The results have been explained.

Round 3

Reviewer 3 Report

I still some concerns about the scientific approach of the article. However, I must recognize that the authors have implemented now some of the most important recommendations made in my previous report.

Consequently, I believe that the article could be considered for publication if the authors delve a little deeper into the question of the proposed case study. Specifically, I think it would be necessary to include at least a couple of figures to make it easier for readers to understand the analysis performed.

First, a presentation figure of the area of study, in which the geographical context to be analyzed is located (the authors have to take into account that Sustainability is an international journal whose readers may have no idea of where is Málaga). The typical location map picture of the city in the national or European context may be sufficient (include at least UTM coordinates and scale).

Secondly, a more concrete image of the study area where there is some small summary of the spatial analysis of the maritime facade of the city. In this case, a map or an orthophoto with a analysis legend that shows more or less some graphic explanation of the analysis performed or the parameters proposed would be very illustrative for the understanding of the interpretation of the indicators carried out.

Author Response

First, a presentation figure of the area of study, in which the geographical context to be analyzed is located. The typical location map picture of the city in the national or European context may be sufficient (include at least UTM coordinates and scale).

Secondly, a more concrete image of the study area where there is some small summary of the spatial analysis of the maritime facade of the city. In this case, a map or an orthophoto with a analysis legend that shows more or less some graphic explanation of the analysis performed or the parameters proposed would be very illustrative for the understanding of the interpretation of the indicators carried out.

A location map of the city in the European context with UTM coordinates has been included (Figure 1).

A metropolitan city map with the main uses and infrastructure has been included (Figure 2).

A graphic explanation of the analysis performed in a study area map has been included (Figure 3).

Thank you very much for your helpful suggestions that provided important inputs for this paper.